# Inhibition of Carbonic Anhydrase 2 Overcomes Temozolomide Resistance in Glioblastoma Cells

**DOI:** 10.3390/ijms23010157

**Published:** 2021-12-23

**Authors:** Kai Zhao, Agnes Schäfer, Zhuo Zhang, Katharina Elsässer, Carsten Culmsee, Li Zhong, Axel Pagenstecher, Christopher Nimsky, Jörg W. Bartsch

**Affiliations:** 1Department of Neurosurgery, Uniklinikum Giessen and Marburg (UKGM), University of Marburg, Baldingerstraße, 35033 Marburg, Germany; Zhouk@students.uni-marburg.de (K.Z.); Schaef4g@students.uni-marburg.de (A.S.); zhzmclaren@gmail.com (Z.Z.); nimsky@med.uni-marburg.de (C.N.); 2Department of Pharmacology and Clinical Pharmacology, Biochemical-Pharmacological Center, University of Marburg, Karl-von-Frisch-Strasse 2, 35032 Marburg, Germany; elsaessk@staff.uni-marburg.de (K.E.); culmsee@staff.uni-marburg.de (C.C.); 3Center for Mind, Brain and Behavior, 35032 Marburg, Germany; pagenste@med.uni-marburg.de; 4College of Bioengineering, Chongqing University, Shazheng Street 174, Shapingba District, Chongqing 400044, China; jlzhong@cqu.edu.cn; 5Department of Neuropathology, Uniklinikum Giessen and Marburg (UKGM), University of Marburg, Baldingerstraße, 35033 Marburg, Germany

**Keywords:** glioblastoma, GBM stem-like cells, carbonic anhydrase 2, temozolomide, chemoresistance, GBM recurrence, acetazolamide, brinzolamide, autophagy

## Abstract

About 95% of Glioblastoma (GBM) patients experience tumor relapse as a consequence of resistance to the first-line standard chemotherapy using temozolomide (TMZ). Recent studies reported consistently elevated expression levels of carbonic anhydrase CA2 in recurrent glioblastoma and temozolomide-resistant glioblastoma stem-like cells (GSCs). Here we show that CA2 is preferentially expressed in GSCs and upregulated by TMZ treatment. When expressed in GBM cell lines, CA2 exerts significant metabolic changes reflected by enhanced oxygen consumption and increased extracellular acidification causing higher rates of cell invasion. Notably, GBM cells expressing CA2 respond to combined treatment with TMZ and brinzolamide (BRZ), a non-toxic and potent CA2 inhibitor. Interestingly, brinzolamide was more effective than the pan-CA inhibitor Acetazolamide (ACZ) to sensitize naïve GSCs and TMZ-resistant GSCs to TMZ induced cell death. Mechanistically, we demonstrated that the combined treatment of GBM stem cells with TMZ and BRZ caused autophagy of GBM cell lines and GSCs, reflected by enhanced LC3 cleavage (LC3-II) and p62 reduction. Our findings illustrate the potential of CA2 as a chemo-sensitizing drug target in recurrent GBM and propose a combined treatment of TMZ with CA2 inhibitor to tackle GBM chemoresistance and recurrence.

## 1. Introduction

Glioblastoma (GBM) is the most common and lethal brain tumor with a median survival of around 15 months after diagnosis [1,2]. Despite an aggressive therapy regimen consisting of surgical resection, adjuvant radiation, and chemotherapy with temozolomide (TMZ) [3] virtually all GBMs recur [4,5]. 

GBM is characterized by a high degree of heterogeneity on phenotypic, genetic, and cellular levels [6]. Like other solid tumors, GBM is composed of various brain resident, as well as transformed cell types: there are rapidly multiplying tumor cells that make up the bulk of the tumor mass and, on the other hand, there are self-renewing cell types, often termed Glioblastoma stem-like cells (GSCs) [6,7,8]. Whereas differentiated GBM cells are responsive to chemotherapy due to their high proliferation rate, GSCs are thought to exert increased resistance to adjuvant therapy and tumor-initiating capacity as a source of glioma recurrence [3,5,9]. There are several proposed resistance mechanisms, such as metabolic inactivation of drugs, inhibition of conversion from prodrug to bioactive drug, increased drug efflux, and increased DNA repair [10,11]. In this regard, by analyzing GSCs resistant to TMZ treatment, we and others have shown previously that a major factor of TMZ resistance in GBM stem cells and recurrent tumor tissue is Carbonic Anhydrase 2 (CA2). More recently, two lines of evidence identified CA2 as another important carbonic anhydrase functionally important for TMZ resistance as a downstream target gene of BCL-3 activation [12,13]. CA2 is a member of a greater family of Carbonic Anhydrases which are highly conserved proteins in animals and plants [14,15,16]. Carbonic Anhydrases constitute a family of structurally divergent proton exchange proteins and particularly CA9 and CA12 have been previously described in GBM and other tumor entities as tumor cell-intrinsic carbonic anhydrases [17,18] and a current clinical trial uses acetazolamide in conjunction with TMZ (trial number NCT03011671) in GBM patients. CA9 has been identified as a hypoxia-dependent gene that is particularly relevant for the stem cell niche, in which GSCs can survive tumor-targeted therapies [19,20]. While CA9 and CA12 are transmembrane proteins, CA2 is located at the inner cell membrane and is associated with monocarboxylate transporter proteins MCT1/4, which are important in the symport of monocarboxylic acids (e.g., lactate, pyruvate) and protons (H^+^) [21]. This supports the notion that CA2 could be involved in proton/lactate symport in GBM cells thereby supporting the so-called “Warburg effect” in tumor cells by maintaining glycolysis as a major source of metabolic energy [22]. 

As no functional data on CA2 in GBM cells are available so far, we analyzed the functional effects of CA2 expression in GBM cells by assessing metabolic parameters, such as the oxygen consumption rate (OCR) and the extracellular acidification rate (ECAR) in cells expressing CA2. Moreover, we investigated how CA2 affects proliferation, invasion, and TMZ resistance. As a potent inhibitor of CA2 with a K_i_ value of 3 nM for the full-length enzyme, we utilized brinzolamide (BRZ) in cell viability assays in comparison to the well-established pan-CA inhibitor acetazolamide (ACZ) and analyzed the potential and the mechanism of CA2 inhibition on TMZ sensitivity. We show that the therapeutic effect of brinzolamide is advantageous compared to acetazolamide. 

## 2. Results

### Expression of CA2 in GBM Patients and GBM Stem-like Cells

A TCGA X GTEx dataset comparing normal brain tissue with that of GBM revealed significantly higher expression levels of CA2 in GBM tissue (Appendix A, *p* < 0.01). Glioblastoma stem-like cells (GSCs) are the major tumor cell type exerting chemoresistance with a strong capacity for self-renewal, one of the main reasons for GBM recurrence. For this reason, we analyzed patient-matched GBM samples from initial and from recurrent GBMs (20 samples from 10 patients). In this particular patient cohort, we observed that CA2 is significantly up-regulated (Figure 1A) while the expression of P-gp, an efflux transport protein is slightly up-regulated, and that of CD133, a GSC marker, is not significantly changed (Figure 1B,C), patient information is shown in Appendix A.

To characterize the CA2 producing cells in recurrent glioblastoma (rGBM), the localization of CA2 was determined by immunofluorescence in paraffin slides of rGBM patients (Figure 1D). In combination with Sox-2, a stem-like cell marker, co-localization with CA2 was observed in representative GBM sections (Figure 1D), suggesting that CA2 is expressed in GSCs of recurrent tumors or differentiated tumor cells (Patient 3, Figure 1D), patient information is shown in Appendix A. Expression levels of CA2 were further investigated in GBM cell lines U87 and U251 and in four patient-derived GSC lines (Figure 1E) and compared to those of CA9 (Figure 1F) and CA12 (Figure 1G), the protein level of CA2 also increased in GSCs (Appendix A). 

Notably, higher expression levels of CA2 mRNA were detected in all GSCs compared to the GBM cell lines, whereas mRNA expression of CA9 was lower in all GSCs, and only one GSC line showed a higher expression level of CA12 while two of them were lower compared to GBM cell lines (Figure 1G). From all three carbonic anhydrase genes, CA9 is the typical gene induced by hypoxia (Appendix A). Based on these observations, the high expression levels of CA2 in GSCs suggest a functional role of CA2. 

To characterize the function of CA2 in GBM cells by gain-of-function analyses, stable cell clones of U87 and U251 GBM cell lines were generated by transfection with either a control vector (U87_Ctrl and U251_Ctrl) or with a CA2 plasmid (U87_CA2 and U251_CA2). After selection with G418, stable cell clones were generated and analyzed for CA2 expression by Western Blot and qPCR (Figure 2A,C), and one representative cell clone from each condition was selected for further analysis. U87_CA2 and U251_CA2 cells gave rise to a 29 kD band of CA2, whereas the control cells were negative for CA2. 

Cell proliferation of these cell clones was determined using a CellTiter-Glo^®^ assay (Figure 2B,D). Both CA2 overexpressing GBM cell types show significantly higher proliferation rates than their respective control clones (Figure 2B,D, with *p* < 0.001 and *p* < 0.01, respectively). In addition, the invasive GBM cell line U251 shows higher rates of cell invasion into Matrigel when expressing CA2 (Figure 2E,F, *p* < 0.001).

To analyze whether CA2 expression affects the cellular metabolism of U87 and U251 cells, we used a Seahorse Analyzer that continuously measures oxygen concentration as oxygen consumption rate (OCR) and the proton flux into the cell supernatant determined as extracellular acidification rate (ECAR) enabling quantification of mitochondrial respiration, ATP production, and glycolysis. Indicative for CA2 function, a notable increase in OCR (Figure 2G,I) and ECAR (Figure 2H,J) was observed in both GBM lines U87 and U251 expressing CA2 compared to their control clones. 

To test if these metabolic changes and the observed invasive behavior can be specifically ascribed to CA2 function, either the pan-CA inhibitor acetazolamide (ACZ, Figure 3A) or the potent CA2 inhibitor brinzolamide (BRZ, Figure 3B) were used in concentrations of 100 and 400 μM, respectively in U251_Ctrl and U251_CA2 cells (Figure 3). The inhibitory effects of ACZ and BRZ clinically used drugs against the carbonic anhydrase isoforms are shown in Appendix A [23,24] and Appendix A (https://www.selleckchem.com/carbonic-anhydrase.html; accessed on 12 December 2021). OCR and ECAR rates were determined after adding either ACZ or BRZ. Metabolic changes were hardly observed in U251_Ctrl cells (Appendix A). In contrast, U251_CA2 cells respond well to the treatment after 24 h with a significantly reduced basal OCR and ECAR (Figure 3C,D), however, to a greater extent with ACZ, but at the higher dose of 400 μM BRZ, cells also responded with a significantly reduced OCR and ECAR. The maximal respiration capacity, after the addition of FCCP, was reduced by 400 μM ACZ (Figure 3E) and 400 μM BRZ (Figure 3G) in U251_CA2 cells compared to U251_Ctrl cells. ACZ and BRZ effectively inhibited the activity of carbonic anhydrase, which decreased cellular respiration, ATP production (Figure 3E, G), and extracellular acidification (Figure 3F,H) in U251_CA2 cells compared to U251_Ctrl cells. However, this effect was more pronounced after ACZ stimulation, which might be based on the fact that ACZ is much more potent than BRZ to modulate mitochondrial activity, since ACZ inhibits CA2 and other CA family members, such as CA9, CA12 which are expressed in both GBM cell lines independent of CA2 expression (Appendix A) and are involved in the mitochondrial metabolism.

To analyze the role of CA2 for invasion behavior of U251 cells, U251_Ctrl cells (Appendix A) and U251_CA2 cells (Figure 3I,J) were analyzed in their response to 100 μM, 400 μM ACZ and 100 μM, 400 μM BRZ stimulation. Whereas ACZ (Figure 3K) had no significant effect on the invasion of U251_CA2 cells, BRZ (Figure 3L) was much more effective in reducing invasion by 10% in a concentration of 400 μM while treatment of U251_Ctrl with ACZ or BRZ did not affect cellular invasiveness (Appendix A). Taken together, these findings indicate that ACZ and BRZ effectively inhibit mitochondrial metabolism with ACZ being more potent than BRZ, whereas BRZ is more effective in reducing the invasiveness of CA2 expressing GBM cells.

To address the major question of whether CA2 inhibition can sensitize GBM cells to TMZ treatment as demonstrated for ACZ [12], we tested the combined application of BRZ with TMZ and hypothesized that CA2 overexpressing U87 and U251 cells can be sensitized to TMZ when CA2 is simultaneously inhibited. We compared the effects of ACZ and BRZ alone or in combination with TMZ on GBM cell lines U87_Ctrl and U87_CA2, and U251_Ctrl and U251_CA2. As previously, two doses of ACZ and BRZ, 100 μM and 400 μM, combined with TMZ (500 μM for U87 and 30 μM for U251) were applied to GBM cell lines for 5 d, and cell viability was determined by a CellTiter-Glo^®^ assay (Figure 4). We observed a synergistic effect of TMZ and BRZ, already at the lower concentration of BRZ on the cell viability of U87_CA2 cells as compared to U87_Ctrl cells, whereas ACZ, combined with TMZ reduced cell viability in U87_CA2 cells than U87_Ctrl cells only at the higher concentration of 400 μM, and there was no change at 100 μM ACZ (Figure 4A). Similarly, BRZ at concentrations of 100 μM and 400 μM combined with TMZ significantly augmented cell death of U251_CA2 cells compared to U251_Ctrl cells (Figure 4C). Interestingly, we also found that monotherapy with TMZ significantly decreased the growth of U87_Ctrl and U251_Ctrl cells compared to U87_CA2 and U251_CA2 cells, respectively, including that U87 and U251 control cells might be more sensitive than CA2 cells in response to TMZ treatment. Concerning toxicity, we show that ACZ alone has no effect on U87_CA2 and U251_CA2 growth, and BRZ alone treatment on U251_CA2 cells also showed no cytotoxic, however, 400 μM BRZ show a slight cytotoxic effect on U87_CA2 cells (Figure 4B,D). 

Together, these data confirm that BRZ in combination with TMZ is more effective than the pan-CA inhibitor ACZ. However, this needs to be demonstrated on GSCs, as this cell type is a major source of TMZ therapy resistance in GBM. Therefore, to analyze the possibility that CA2 induced by TMZ could cause chemoresistance, we first evaluated the mRNA expression of CA2 in three independent patient-derived GSC lines after stimulation with TMZ and determined CA2 expression levels by qPCR (Figure 5A–C). After stimulating the GSCs with 500 μM TMZ for 3 d and 5 d, expression of CA2 was found to be significantly up-regulated in GSC151 cells after 3 and 5 days, in GSC175 cells only after 5 days and GSC 240 cells 3 days after TMZ treatment (Figure 5A–C). 

Considering that CA2 expression is induced by TMZ stimulation in GSCs, inhibition of CA2 expression might overcome TMZ resistance. To confirm our hypothesis, we determined cell viability using a CellTiter-Glo^®^ assay after treatment with TMZ, ACZ, BRZ alone or in combination for 10 days (Figure 5D–G). Visualization of the morphology of GSCs using light microscopy identified differences in cell growth with TMZ alone, ACZ/BRZ alone, or combined ACZ/BRZ plus TMZ treatment (Figure 5D). 

Our quantitative results show that TMZ alone diminishes cell numbers by more than 50% as compared to the control group in all three GSCs, whereas ACZ alone or BRZ alone did not have any cytotoxic effect on GSCs, even at higher concentrations of 400 μM. The combination of TMZ with ACZ at 100 μM did not cause growth inhibition compared to TMZ alone, whereas co-treatment with 400 μM ACZ and TMZ significantly reduced cell viability than single treatment with TMZ. However, BRZ, at both concentrations (100 μM and 400 μM) significantly augments TMZ cytotoxicity in all GSC lines. These results indicate that co-treatment with the CA inhibitor BRZ and TMZ leads to a better re-sensitization of GBM stem cells than ACZ applied in combination with TMZ (Figure 5E–G). 

Next, we sought to explore the cytotoxic effects of single TMZ treatment and co-treatment with ACZ or BRZ in a GSC line with a high TMZ resistance (GSC_TMZ), resembling a GSC type present in recurrent GBM in comparison to the naïve GSC line (GSC_DMSO) [12]. Firstly, to explore the changes in mRNA expression levels of carbonic anhydrases CA2, CA9, CA12 (Figure 6A–C), monocarboxylate transporter molecules MCT1, MCT4, SLC4A4 (Figure 6D–F), and drug-resistant protein molecule (P-gp, Figure 6G) as a consequence of long-term treatment with TMZ and acquired chemoresistance, the lines GSC175_Ctrl, CSC175_DMSO, and GSC175_TMZ resistant GSCs were analyzed by q PCR (Figure 6A–G). We noticed that GSC_TMZ cells significantly up-regulated CA2, CA12, P-gp, SLC4A4 expression levels compared to GSC_DMSO or GSC_Ctrl cells, while MCT1 and MCT4 mRNA transcripts were not changed, even CA9 mRNA expression was reduced in GSC_TMZ cells, confirming that the activation of CA2 and CA12 might be involved in TMZ-mediated GSC resistance, meanwhile, Western blot also shown an upregulation of CA2 in GSC_TMZ cell (Appendix A). On the contrary, GSC_TMZ did not up-regulate CA9, MCT1, MCT4 mRNA expression, indicating that these proteins mediating intracellular pH and lactate transport may require a hypoxic microenvironment in glioblastomas. To test for the viability of these cells, they were observed under the light microscope after 14 d stimulated with ACZ, BRZ alone, and in combination with TMZ (Figure 6H). Here, cells were treated with the IC_50_ value of TMZ as shown previously [12]. Similar to the reported combined treatment regimens, 100 μM and 400 μM of CA inhibitors ACZ and BRZ were applied. As expected, single TMZ treatment almost reduced half of the cells compared to the control group, which was significant in GSC_DMSO cells and GSC_TMZ cells (Figure 6I,J). Neither 100 μM nor 400 μM ACZ or BRZ alone caused any cytotoxicity in these cells. Noticeably, the combined treatment of ACZ with TMZ did not have any effect on cell viability compared to TMZ treatment alone, either in GSC_DMSO cells or in GSC_TMZ cells (Figure 6I,J). However, co-treatment with BRZ was more cytotoxic at lower concentrations in GSC_TMZ cells than GSC_DMSO cells and significantly inhibited cell viability (Figure 6I,J), which might be due to CA2 overexpression in GSC_TMZ cells, while BRZ more specifically inhibits CA2 expression than ACZ. Collectively, these data indicated that BRZ combinatorial treatment might be most potent in long-term TMZ-resistant GSCs and that BRZ is synergistic to TMZ to cause increased cytotoxicity of TMZ to reduce chemoresistance of GSCs. 

As established previously, autophagy is altered in GBM cells and can be induced by TMZ, so that many resistance mechanisms exist to prevent chemotherapy-induced autophagy. Autophagy itself might exert either a pro-tumor or an anti-tumor effect. We, therefore, asked whether this phenomenon that we observed above might be due to co-treatment with CA inhibitors promoting cell death via enhanced activation of autophagy. To test our hypothesis, LC3II, the lipidated autophagosome-associated form of LC3, and p62, a marker of autophagic flux were measured by Western Blot and as LC3-II puncta formation by immunocytochemistry. We noticed that BRZ/TMZ but not ACZ or BRZ enhanced autophagy of U251_CA2 cells (ratio of LC3II to LC3I) (Figure 7A,B). 

We also examined the effects of the combined treatment with TMZ/ACZ or TMZ/BRZ on CA2, LC3, p62 expression in U87_CA2 and U251_CA2 cells (Appendix A and Figure 7). Our results show that TMZ significantly up-regulates the protein level of CA2 while we found that co-treatment with BRZ effectively inhibited CA2 expression compared to TMZ, BRZ, ACZ alone, or to combined TMZ/ACZ treatment (Figure 7D–G), which demonstrated that BRZ more specifically inhibits CA2 expression than ACZ after TMZ stimulation. Moreover, BRZ + TMZ significantly increased the conversion of LC3I to LC3II compared with TMZ alone or ACZ + TMZ, and also decreased p62 expression. The LC3 and p62 expression in U251_Ctrl cells as well as U87_Ctrl cells enhanced autophagy by single TMZ treatment but not by co-treatment (Figure 7C and Appendix A) suggesting that CA2 is required to exert this effect. For U251_CA2 cells, we observed that TMZ in combination with BRZ significantly increased the LC3II/LC3I ratio and reduced CA2 and p62 expression (Figure 7D), which suggests that BRZ in combination with TMZ strongly increases autophagic flux and promotes the occurrence of autophagy thereby causing enhanced cell death. 

These results indicate that BRZ might be involved in mediating TMZ chemosensitivity of CA2 overexpressing GBM cell lines through enhanced activation of autophagy. This mechanism is similar in GSCs, as the combination of BRZ and TMZ also increased cell death of GSCs. For a precise evaluation of the autophagic progress in GSCs, CA2, LC3, p62 were measured by Western Blot after treatment with TMZ alone, ACZ/BRZ alone, or co-treatment for 24 h and compared to controls. All three GSCs showed, in the presence of BRZ, a significant increase in LC3II that was accompanied by a decrease in LC3I in Western Blots (Figure 7E–G). This was even more pronounced in cells treated with TMZ/BRZ. We also found that p62 was reduced under TMZ/BRZ co-treatment. Furthermore, CA2 was significantly decreased in TMZ combined with BRZ in all three GSCs. These patterns are consistent with an increase in autophagy.

## 3. Discussion

The current standard strategies for patients with newly diagnosed GBM glioma include surgical resection, radiation therapy, and chemotherapy [3]. However, in most patients, tumor recurrence is an inevitable problem. Singh et al. first demonstrated the existence of glioma stem-like cells (GSCs) in human brain tumors [25,26]. Further research established that GSCs play important roles in therapeutic resistance, including invasion [27], chemo-resistance [28], radio-resistance [11], and recurrence [28]. As general properties of all stem cells, a GSC has the potential for self-renewal, a capacity for undirected differentiation and tumor initiation, and GSC chemo-resistance is related to drug efflux transporter and GSC diversity [29,30], GSCs have been identified to be relatively resistant to the first-line chemotherapeutic drug TMZ compared to their non-stem-like counterparts [31]. Glioma GSCs have higher expression levels of O6-methyl-guanine- DNA-methyltransferase, the key repair enzyme for TMZ-induced DNA damage, compared to non-stem cells [32]. Therefore, improving the efficacy of TMZ to overcome GSCs resistance is an urgent problem that needs to be solved.

In our study, we identified a strategy to overcome GBM chemoresistance by pharmacological targeting of carbonic anhydrase 2 (CA2) for the improvement of TMZ efficacy which could have implications for further clinical strategies in GBM treatment. Transcriptomic data from recurrent tumor tissues and TMZ-resistant GSCs previously revealed consistently enhanced levels of CA2, however, no functional data on CA2 in GBM cells and GSCs were available. Firstly, we found higher expression levels of CA2 in GSCs compared to GBM cell lines, U87 and U251. To investigate the functional consequences of CA2 expression in these cells, we established stable CA2 overexpressing cell clones of U87 and U251 cells and determined their cellular behavior. 

As evolutionary highly conserved zinc-dependent proteins with one of the highest substrate turnover rates, CAs catalyze the conversion of CO_2_ and H_2_O into bicarbonate. Hence, CAs are involved in cellular processes, such as mitochondrial respiration and acid-base regulation, electrolyte secretion, bone reabsorption, and calcification, also in lipogenesis and gluconeogenesis [33]. As a highly active cytosolic isoform, CA2 is expressed in almost all tissues throughout the body including the brain [34,35]. It was shown that CA2 interacts with monocarboxylate transporter proteins e.g., MCT-1/4 thereby symporting protons with monocarboxylic acids, such as lactate [21]. As a metabolic consequence, CA2 maintains glycolysis in tumor cells, thus supporting the “Warburg effect”, prioritizing glycolysis as a major source of energy. Interestingly, it has been proposed that CA2 interacts with V-ATPase [36] to control intracellular energy flow and cellular autophagy. Our data demonstrate that, in GBM cells, high expression of CA2 leads to an increase in mitochondrial basal respiration, maximal respiration, ATP production, and glycolytic activity that these alterations may link to patient outcome, this finding allows us to further clarify the role of CA2 in GBM. 

Acetazolamide (ACZ) has been reported to have synergistic effects when combined with TMZ [13,37,38] and is currently in clinical studies for GBM co-treatment with TMZ (https:clinicaltrials.gov; Study Number NCT03011671; accessed on 9 November 2021) in a dose of 500 mg per day and escalated to 1000 mg per day between cycle day 1–21. However, due to its pan-CA characteristics, ACZ could inhibit carbonic anhydrases which could have beneficial effects, i.e., by sensitizing GBM cells to TMZ, so that the ACZ effect might be less significant as more specific CA2 inhibition. Indeed, depending on the cell type investigated, there is a difference in the efficacy of BRZ/TMZ treatment compared to ACZ/TMZ treatment, which is highly significant in U251_CA2 cells and two of the three patient-derived GSC lines. Given these results, we hypothesize that a combination of the CA2 inhibitor brinzolamide (BRZ) in conjunction with TMZ might provide an improved patient benefit by acting on several cellular processes in which carbonic anhydrases are involved. 

Concerning carbonic anhydrases known to be physiologically relevant, the specificity of CA-inhibitors needs to be evaluated (Appendix A). In contrast to acetazolamide, brinzolamide is considered to be the most specific CA2 inhibitor with a K_i_ value of 3 nM for the full-length enzyme. However, brinzolamide is also able to inhibit CA12 with a K_i_ value of 3 nM for the catalytic domain. As an argument for higher specificity of brinzolamide for CA2 over CA12, the K_i_ value for the full-length enzyme of CA12 is unknown and K_i_ values for the full-length enzyme are likely by an order of magnitude higher than for the isolated catalytic domain [23,24]. Nevertheless, we can state an improved therapeutic effect of brinzolamide compared to acetazolamide when combined with TMZ.

In this respect, our study demonstrates the differences between the application of a pan-CA and a more specific CA2 inhibitor. Metabolic processes in GSCs were more severely impaired by ACZ as compared to BRZ. This can be explained by the fact that all CAs present in GBM cells (at least CA2, CA9, and CA12) are part of the inside/out regulation of protons and the monocarboxylate transport mechanism. Thus, as observed in GBM cells, ACZ reduces OCR and ECAE in CA2 overexpressing cells, but the effects of ACZ are significantly stronger than those of BRZ.

However, there might be a certain “hierarchy” in CAs, in which CA2 might have stronger effects on cellular processes apart from metabolism, and this is the induced invasion seen under CA2 expression. It is, therefore, interesting to notice that in this particular cellular process, the effect of BRZ is stronger than that of ACZ. 

Both CA inhibitors, ACZ and BRZ, are not cytotoxic as our data suggest in all cell lines investigated. However, both inhibitors can increase TMZ efficacy. The highest TMZ efficacy is achieved at physiological pH. However, tumor cells often exhibit an alkaline pH, so that acidification as a result of CA2 inhibition leads to a physiological pH, thereby enhancing the uptake of TMZ into tumor cells. The described alterations of metabolism in vivo often causally correlate with hypoxia [13,38]. Hypoxia is a characteristic feature of the GBM microenvironment and impacts several processes which lead to the progression of tumors, such as differentiation, invasion, and angiogenesis [39]. It was shown that TMZ induces autophagy in GBM cells, and additional compounds leading to augmented TMZ-induced autophagy are of great value for the optimization of GBM chemotherapy strategies [30,40]. 

In our study, we confirmed that in GBM cells expressing CA2 and in GSCs, CA2 can suppress autophagy, as autophagy is activated after treatment with the CA inhibitor BRZ [41,42]. Similar, although weaker effects were observed with ACZ. The metabolic changes in ACZ treated cells are more pronounced compared to BRZ, and yet, the effects of BRZ are stronger for invasion behavior and autophagy regulation. The somewhat surprising outcome of this study was that a preferential CA2 inhibition, in contrast to combined inhibition of GBM-relevant CAs (CA2, CA9 and CA12) with ACZ provides a stronger effect on cell viability when combined with TMZ. The strongest effect of BRZ over ACZ was observed for autophagy regulation in CA2 expressing U251 cells and in GSCs. Both CA inhibitors can induce autophagy, and in some cell lines, BRZ was the stronger inducer. However, when TMZ induced the so-called chemotherapy-induced autophagy that leads to cell death, the combined treatment with BRZ showed significantly stronger effects on cell viability than the combination of TMZ with ACZ suggesting that inhibition of other CAs apart from CA2 could cause compensatory effects that can support cell survival. 

In addition to autophagy regulation, we show in our study that inhibition of CA2 by brinzolamide has a strong effect on the invasive behavior of GBM cells which is superior to ACZ. Whether this effect is solely based on metabolic effects remains to be determined.

These findings have implications for clinical studies. Firstly, we show that ACZ is only effective in higher doses of 400 μM. As an estimate, the ACZ concentration of current clinical studies is in the range of 100 μM so that autophagy induction is most likely not sufficient to exert a significant clinical benefit to patients, as these ACZ doses used for co-treatment with TMZ are shown to be inefficient in our cell-based studies as demonstrated here. Based on these data, a TMZ/BRZ co-treatment would be a more promising approach. However, so far, pharmacological use of BRZ is only approved for ophthalmologic applications [43] and it remains to be established whether it can be applied to patients systemically. 

## 4. Materials and Methods 

### 4.1. Clinical Specimens

All tumor tissue samples from patients obtained approval from the ethics committee of the Medical Faculty, Philipps University of Marburg (Institutional review board number 185/11).

### 4.2. Gene Expression Analysis

We checked the CA2 gene expression in the “Gene-DE” part of the TIMER2 website and observed the CA2 difference expression between different tumor tissues and adjacent normal tissues in the TCGA database (http://timer.cistrome.org/; 12 December 2021), data are not shown. For GBM only 5 normal tissues, we used the “Expression analysis Box Plots” part of the GEPIA2 website (http://gepia2.cancer-pku.cn/#analysis, 12 December 2021) to obtain the CA2 expression difference between GBM tumor tissues and the normal tissues of the GETx (Genotype-Tissue Expression) database.

### 4.3. Cell Culture

U87 and U251 glioblastoma cell lines (identity of cell lines was verified by karyotyping and STR profiling) were obtained from ECACC and were cultured in DMEM cell medium (DMEM-HA, Capricorn Scientific, Ebsdorfergrund, Germany) supplemented with 10% fetal bovine serum (FBS) (S0615, Sigma, Dreieich, Germany), 1% penicillin/streptomycin (2321115, Gibco, Carlsbad, CA, USA), 1% Non-essential amino acids (NEAA) (11140050, Gibco, Carlsbad, USA) and 1% Sodium pyruvate (NPY-B, Capricorn Scientific, Ebsdorfergrund, Germany). Glioblastoma stem-like cells were isolated as described previously [12]. Cells were cultured in DMEM/F12 medium (DMEM-12-A, Capricorn Scientific, Ebsdorfergrund, Germany) including 2% B27 supplement (117504044, Gibco, Carlsbad, USA), 1% amphotericin (152290026, Gibco, Carlsbad, USA), 0.5% HEPES (H0887, Sigma, Dreieich, Germany) and 0.1% gentamycin (A2712, Biochrom, Berlin, Germany) with EGF (100-18B, Peprotech, Hamburg, Germany) and bFGF (315-09, Peprotech, Hamburg, Germany) in a concentration of 20 ng/mL, respectively. All cells were incubated at 37 °C under 5% CO_2_ humidified incubator.

### 4.4. Generation of CA2 Expressing Stable GBM Cell Lines 

U87 and U251 CA2 expressing cells were generated by transfection with CA2 plasmid (RC201974, ORIGENE, Rockville, MD, USA) using 2.5 μg of plasmid DNA mix Lipofectamine LTX Reagent (15338100, Invitrogen, Waltham, MA, USA). After 48 h, the medium was changed to complete DMEM with G418 (A2912, Biochrom, Berlin, Germany) (400 μg/mL for U87 cells and 600 μg/mL for U251 cells) for 3 weeks. Positive clones were picked and further selected for stable integration. The plasmid pCMV6-LacZ-Bbsl (114,671, Addgene, Watertown, MA, USA) was used for vector control cell lines.

### 4.5. Invasion Assay

Acetazolamide (ACZ, A6011, Sigma, Dreieich, Germany) and Brinzolamide (BRZ, SML0216, Sigma, Dreieich, Germany) were used for cell treatment 24 h before cell invasiveness analysis. An 8 μm pore transwell insert (662638, Greiner Bio-One, Monroe, Austria) in conjunction with a 24-well plate was used. Transwells were coated with 50 μL matrigel Mix (matrigel mixed 1:1 with cold DMEM) (Corning, 354230, Corning, NY, USA), transwell was turned upside down after 1 h after solidification, and 2.5 × 10^3^ cells in 50 μL medium were seeded on the other side of the insert, turn the transwell right side up again after 4 h adherence. Around 250 μL medium containing 20% FBS was supplied to the transwell on the upper chamber whereas 750 μL medium containing 0.5% FBS was added into the 24-well plate to create a concentration gradient. Cells invaded along the FBS gradient into the matrigel for 24 h, and cells were fixed with 4% PFA, permeabilized with 0.3% Triton (T8787, Sigma, Dreieich, Germany), and nuclei were stained with Hoechst 33342 (1:10,000 dilution, 62249, ThermoFisher, Waltham, MA, USA) for 15 min at RT in the dark. Images were acquired using Leica confocal microscope. Cells were counted in six random areas and were calculated as the percentage of cells that passed across the membrane.

### 4.6. Cell Viability Assay

The survival effect of cells was determined using the CellTiter-Glo 3D cell viability assay (G7571, Promega, Walldorf, Germany). GBM cell lines were seeded at a density of 2.0 × 10^3^ cells per well, GSCs were seeded at a density of 1.0 × 10^4^ cells per well, TMZ-resistant GSCs were seeded as described previously [12]. All cells were stimulated by overnight incubation. 50 μL CellTiter-Glo 3D Reagent was added to the well before measurement, mixed shaking for 15 min, and incubated for 15 min at RT to avoid light. Luminescence was measured with a Microplate Reader luminometer (FLUOstar OPTIMA Microplate Reader, Offenburg, Germany).

### 4.7. Seahorse Assay

The seahorse system XF96-Analyzer (Agilent Technologies, Waldbronn, Germany) was used to measure alterations in the metabolism of cells to analyze mitochondrial oxygen consumption rate (OCR) and extracellular acidification rate (ECAR), 9000 cells were seeded in micro-plate overnight, after inhibitor treatment 24 h, the conditional medium was replaced by seahorse medium and incubated for 1 h before measurement. The whole detailed steps were executed as described before [44]

### 4.8. RNA Isolation, Reverse Transcription, and Quantitative Real-Time PCR

Total RNA was isolated by QIAzol reagent (79306, Qiagen, Hilden, Germany) and absorbance was measured with OD 260/280 ratio between 1.8 and 2.1. For qPCR analysis, 2 μg of RNA was subjected to the synthesis of cDNA by using RNA to cDNA EcoDry Premix (Clontech, Saint-Germain-en-Laye, France) according to the manufacturer’s instructions. Total reaction volume (20 μL) includes 10 μL SYBR Green/Rox Master Mix (Primer Design, Southhampton, UK), 2 μL primers synthesized by Qiagen GmbH (Hilden, Germany), 6 μL nuclease-free water, and 2 μL cDNA. The qPCR protocol set initial denaturation at 95 °C for 10 min, then 40 amplification cycles including 95 °C for 15 s and 60 °C for 1 min. The XS-13 was used as an internal reference for all reactions. The fold changes in gene expression relative to control were calculated by 2^−ΔΔCT^. 

### 4.9. Protein Extraction and Western Blot Analysis

After cells were washed 3 times with ice-cold PBS, total proteins were extracted with RIPA buffer including a protease inhibitor (A32955, Thermo Scientific, Waltham, MA, USA) and phosphatase inhibitor (A32957, Thermo Scientific, Waltham, MA, USA). Protein lysates were boiled in Laemmli and sample reducing buffer (B0009, Invitrogen, Waltham, MA, USA) for 5 min. An equal amount of protein samples were separated by 12.5% SDS polyacrylamide gel electrophoresis and transferred onto nitrocellulose (NC) membranes (A29591442, GE Healthcare Life science, Solingen, Germany). After blocking with 5% non-fat milk (X968.1, Carl Roth, Karlsruhe, Germany) for 1 h at RT, and membrane were incubated overnight at 4 °C with primary antibody, the following antibodies were used: anti-CA2 (1:2000 dilution, ab124687, Abcam, Cambridge, UK), anti-MAP LC3 (1:500 dilution, sc-271625, Santa Cruz Biotechnology, Dallas, TX, USA), anti-p62 (2 ug/mL dilution, MAB8028, R&D Systems, Minneapolis, MN, USA), and anti-β-Tubulin (1:2000 dilution, NB600-936, Novus Biologicals, Littleton, CO, USA) in 5% milk in TBST. Subsequently, the NC membranes were incubated with secondary antibody Donkey (Dnk) pAb to Mouse (Ms) IgG (HRP) (ab97030, Abcam, Cambridge, UK) and Dnk pAb to Rabbit (Rb) IgG (HRP) (ab97064, Abcam, Cambridge, UK) for 1 h at RT. The membranes were washed with TBS-T and scanned with Chemostar Imager (Intas, Goettingen, Germany).

### 4.10. Immunofluorescence Staining

Paraffin sections (5 μm thick) were obtained from GBM patients. All slides were blocked to avoid non-specific binding with 1% BSA (A7030, Sigma, Dreieich, Germany), and incubated with Rabbit Anti-CA2 antibody (1:250 dilution, ab124687, Abcam, Cambridge, UK) and Mouse Anti-Sox2 antibody (1:100 dilution, MA1-014, ThermoFisher, Waltham, MA, USA) at 4 °C overnight, washed in PBS 3 times, and incubated with secondary antibody Donkey anti-Rabbit 550 (1:200 dilution, ab98489, Abcam, Cambridge, UK), Donkey anti-Mouse 488 (1:200 dilution, ab98794, Abcam, Cambridge, UK) at least 1 h at RT avoid night. Nuclei were stained with Hoechst 33342 (1:10,000 dilution, 62249, ThermoFisher, Waltham, MA, USA) 15 min at RT avoid night before cover with anti-fade mounting medium (S3023, Agilent, Santa Clara, CA, USA) was done. Images were acquired using a Leica DM 5500 microscope.

For IF, cells were cultured on collagen I (C7661, Sigma, Dreieich, Germany) coated coverslips in a 24-well plate overnight, cells were washed in PBS 3X, fixed with 4% paraformaldehyde 15 min at RT, permeabilized with 0.3% Triton X100 (T8787, Sigma, Dreieich, Germany) 15 min at RT, blocked to avoid non-specific binding with 5% BSA 1 h, and incubated with Mouse anti-MAP LC3 (1:250 dilution, sc-271625, Santa Cruz Biotechnology, Dallas, USA) at 4 °C overnight, washed, and incubated with secondary antibody donkey anti-Mouse 488 (1:200 dilution, ab98794, Abcam, Cambridge, UK) at least 1 h at RT avoid night. Nuclei were stained with Hoechst 33342 15 min at RT in the dark before covering with the anti-fade mounting medium was done.

### 4.11. Statistical Analyses 

All data were shown as the mean ± SD or SEM and analyzed using GraphPad Prism software (GraphPad Software Inc., San Diego, CA, USA). Unpaired Student’s *t*-tests were used for statistical comparison among two groups. Analysis of variance (ANOVA) test was performed for multicomponent comparisons (one-way or two-way ANOVA), all statistical tests have been indicated in the figure legends. *p*-value < 0.05 was considered statistically significant.

## 5. Conclusions

As recurrence caused by TMZ resistance is almost unavoidable for GBM patients, there is an urgent need for improved treatment options. Here we show the functional consequences of CA2 in GBM cell lines and GSCs and provide a rationale for inhibition of CA2 bearingf a novel therapeutic potential by sensitizing GBM cell lines and GSCs for TMZ treatment more efficiently as pan-CA inhibition using acetazolamide. 

## Figures and Tables

**Figure 1 ijms-23-00157-f001:**
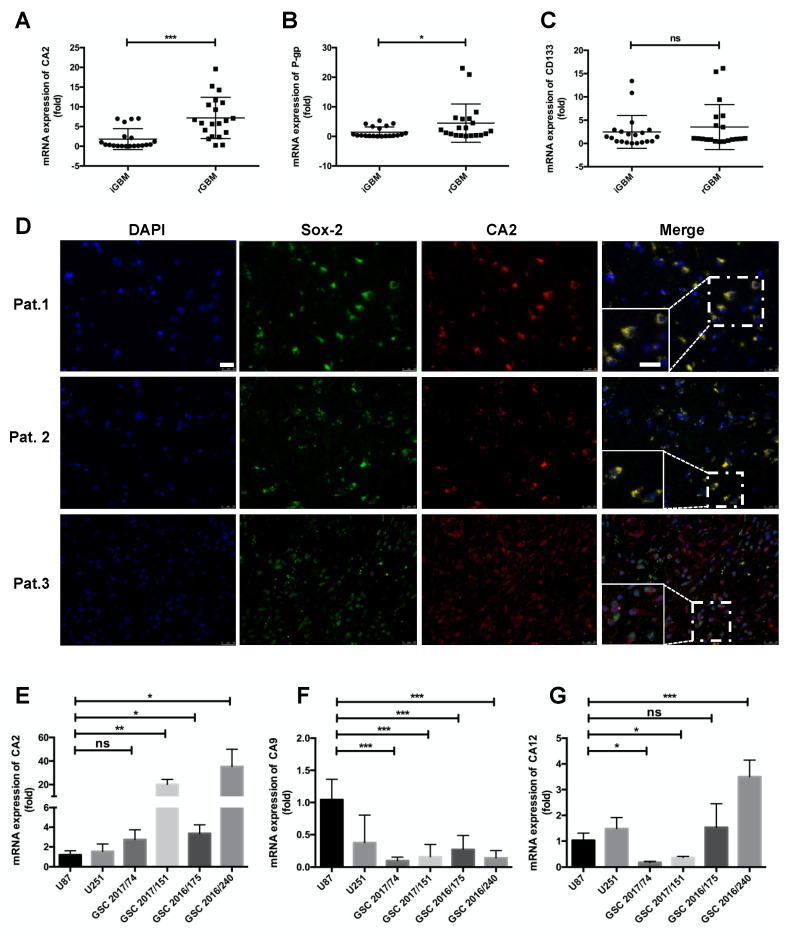
Expression levels of CA2 (**A**), P-gp (**B**), and CD133 (**C**) in patient-matched initial (i) vs. recurrent (r) GBM (*n* = 20 from 10 matched patients); (**D**) Representative IF staining of CA2 and stem cell marker SOX2 in three GBM tissue sections. Co-staining of CA2 (red) and SOX2 (green) confirmed CA2 in stem-like cells of GBM patients (*n* = 3 from 10 recurrent GBM patients), scale bar: 25 μm. (**E**–**G**) mRNA expression of GBM related carbonic anhydrase genes *CA2* (**E**), *CA9* (**F**), and *CA12* (**G**) in GBM cell lines U87 and U251 and patient-derived GSCs. Carbonic Anhydrase expression levels were detected by qPCR (*n* = 3). For all genes, expression levels determined in U87 cells were set to 1. qPCR results were obtained from three independent experiments. In (**A**–**C**,**E**–**G**), data are presented as mean ± SD, student’s *t*-test was used to analyze (**A**–**C**), One-way ANOVA was used to analyze (**E**–**G**), * *p* < 0.05; ** *p* < 0.01; *** *p* < 0.001, ns: not significant.

**Figure 2 ijms-23-00157-f002:**
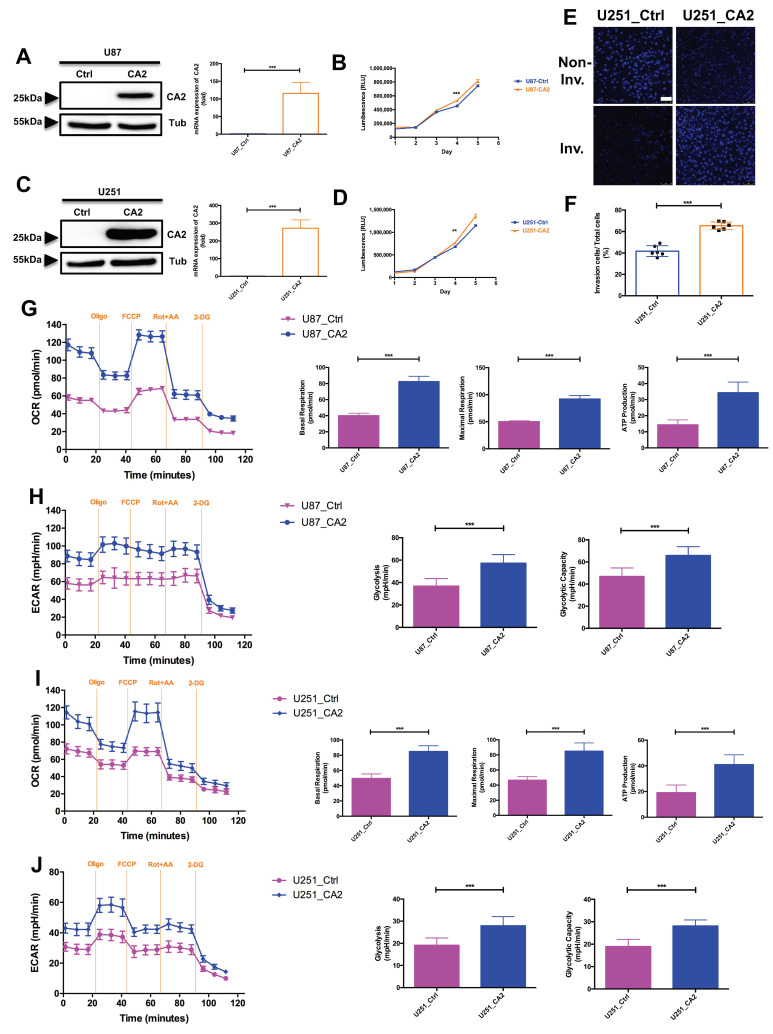
(**A**,**C**) Generation of stable CA2 overexpressing U87 and U251 cell lines as confirmed by Western Blot and qPCR. (**B**,**D**) The proliferation of control cells (U87_Ctrl, U251_Ctrl) and CA2 overexpressing (U87_CA2, U251_CA2) cells was measured by CellTiter Glo (*n* = 3 independent replicates). (**E**) Invasion of U251_Ctrl (Left panel) and U251_CA2 (Right panel) cells stained with DAPI (scale bar: 75 μm). (**F**) Relative invasion of U251_Ctrl and U251_CA2 cells (*n* = 6). (**G**,**I**) CA2 overexpressing cells have increased oxidative mitochondrial metabolism. The oxygen consumption rate (OCR) was measured by a seahorse XFe96 metabolic-flux analyzer. U87_CA2 cells (**G**) and U251_CA2 cells (**I**) significantly increased mitochondrial basal respiration, maximal respiration, and ATP production (*n* = 7–8). (**H**,**J**) CA2 overexpressing cells have elevated levels of glycolysis rate. The extracellular acidification rates (ECAR) were measured by a seahorse XFe96 metabolic-flux analyzer. U87_CA2 cells (**H**) and U251_CA2 cells (**J**) significantly increased glycolytic activity (*n* = 7–8). Results were obtained from three independent experiments. Data are presented as mean ± SD, student’s *t*-test was used to analyze the data ** *p* < 0.01; *** *p* < 0.001, ns: not significant.

**Figure 3 ijms-23-00157-f003:**
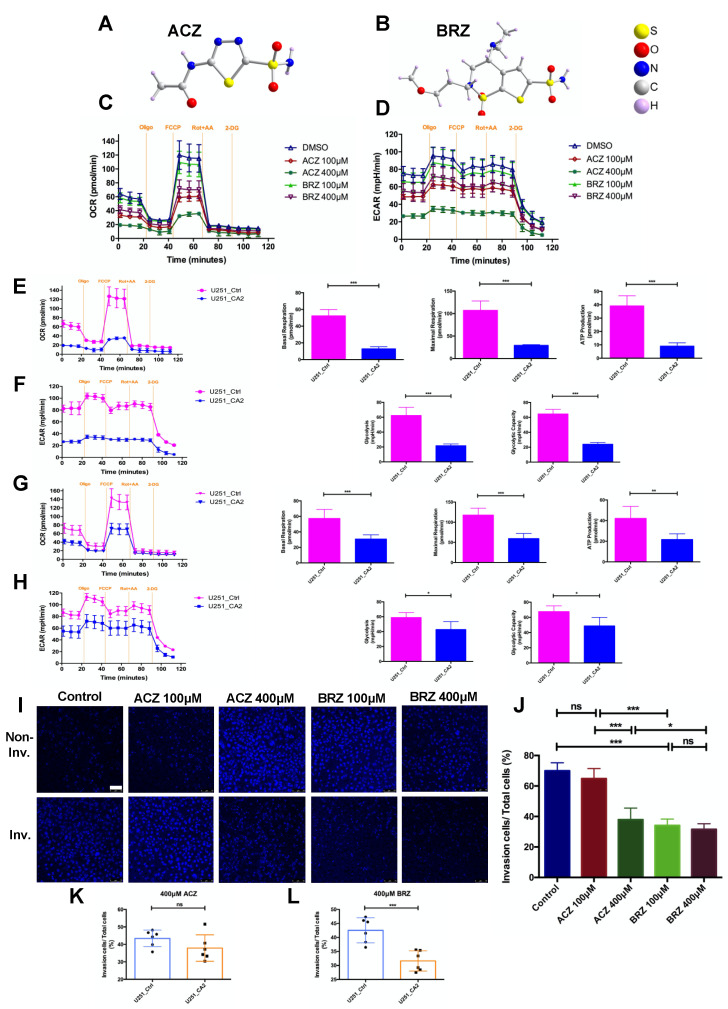
Effect of the pan-CA inhibitor acetazolamide (ACZ) and a potent CA2 inhibitor brinzolamide (BRZ) on invasion and energy metabolism in GBM cell lines. (**A**,**B**) Molecular structures of ACZ and BRZ. (**C**,**D**) U251_CA2 cells’ overall metabolism after ACZ and BRZ stimulation was measured by a seahorse XFe96 metabolic-flux analyzer. ACZ and BRZ significantly decreased oxidative metabolism (**C**) and also reduced the level of glycolysis rate (**D**) in U251_CA2 cells (*n* = 5–6). U251_CA2 cells significantly decreased mitochondrial basal respiration, maximal respiration, and ATP production compared to U251_Ctrl cells after a 24 h stimulation with either 400 μM ACZ (**E**) or 400 μM BRZ (**G**). U251_CA2 cells also significantly reduced glycolytic activity compared to U251_Ctrl cells after 400 μM ACZ (**F**) and BRZ (**H**) stimulation for 24 h. (**I**) Invasion assay of U251_CA2 cells stained with DAPI after ACZ and BRZ stimulation for 24 h with concentrations indicated (scale bar: 75 μm). (**J**) Quantification of the proportion of invasive cells. Both 100 μM and 400 μM BRZ significantly decreased cell invasion of U251_CA2 cells, whereas only 400 μM ACZ reduced the invasiveness of cells (*n* = 6). (**K**,**L**) U251_CA2 cell significantly reduced cell invasiveness compared to U251_Ctrl cells after treatment with 400 μM BRZ. However, 400 μM ACZ did not change the invasiveness of U251_CA2 cells compared to the control group. Results were obtained from three independent experiments. Data are presented as mean ± SD, student’s *t*-test was used to analyze (**E**–**H**,**K**,**L**) One-way ANOVA was used to analyze (**J**), * *p* < 0.05; ** *p* < 0.01; *** *p* < 0.001, ns: not significant.

**Figure 4 ijms-23-00157-f004:**
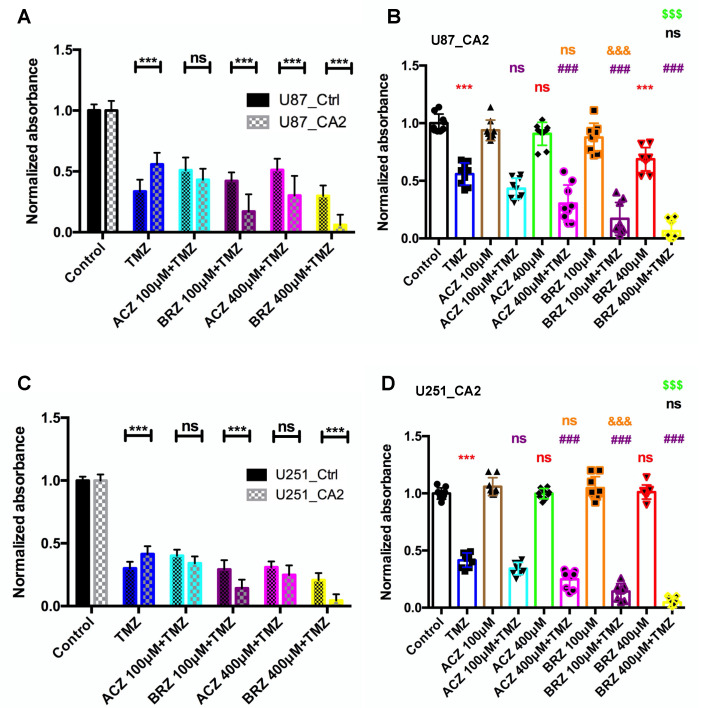
Effect of combined treatment with TMZ and either ACZ or BRZ on CA2 overexpressing GBM cell lines U87 and U251. (**A**) Co-treatment with 500 μM TMZ and ACZ/BRZ on U87_Ctrl cells and U87_CA2 cells for 5 d. U87_CA2 cells significantly decreased cell viability compared to U87_Ctrl cells after BRZ + TMZ treatment. However, only 400 μM ACZ + TMZ decreased cell viability compared to the U87_Ctrl group (*n* = 3). (**B**) In U87_CA2 cells treated with TMZ, ACZ/BRZ, or TMZ + ACZ/BRZ, TMZ plus 400 μM ACZ decreased the viability compared to TMZ alone. However, 100 μM ACZ in combination with TMZ did not change the cell viability. Both 100 μM and 400 μM BRZ in combination with 500 μM TMZ decreased the viability significantly compared to TMZ treatment alone (*n* = 3). (**C**) Co-treatment with 30 μM TMZ and ACZ/BRZ on U251_Ctrl cells and U251_CA2 cells for 5 d. U251_CA2 cells decreased cell viability compared to U251_Ctrl cells after 400 μM BRZ + TMZ treatment. However, ACZ + TMZ did not change cell viability compared to the U251_Ctrl group (*n* = 3). (**D**) In U251_CA2 cells, ACZ/BRZ has no cytotoxic effect, TMZ plus 400 μM ACZ decreased the viability compared to TMZ alone. Both 100 μM and 400 μM BRZ in combination with 500 μM TMZ decreased the viability significantly compared to TMZ treatment alone (*n* = 3). * (Red): compared to Control group; # (Purple): compared to TMZ group; & (Orange): compared to ACZ 100 μM + TMZ group; % (Black): compared to BRZ 100 μM + TMZ group; $ (Green): compared to ACZ 400 μM + TMZ group. Results were obtained from 3 independent experiments. Data are presented as mean ± SD, Two-way ANOVA was used to analyze (**A**,**C**), One-way ANOVA was used to analyze (**B**,**D**), ***, $$$, &&&, ###, *p* < 0.001, ns: not significant.

**Figure 5 ijms-23-00157-f005:**
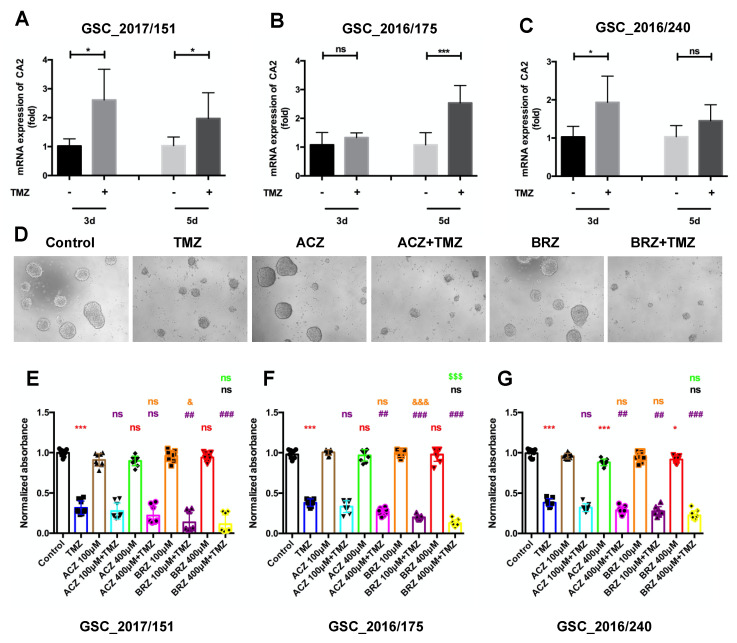
Combined treatment of BRZ and TMZ increases cell death in GBM stem-like cells. (**A**–**C**) mRNA expression of CA2 after TMZ treatment at 3 d and 5 d were detected by RT-PCR, the mRNA level of CA2 increased after TMZ stimulation (*n* = 3). (**D**) Visualization of the morphology of GSCs (Nr. 2016/175) using light microscopy after TMZ and CA inhibitor stimulation for 10 d. (**E**–**G**) Cell viability of co-treatment measured by CellTiter-Glo assay. Co-treatment with 500 μM TMZ and 400 μM ACZ decreased the viability compared to TMZ alone. However, 100 μM ACZ in combination with TMZ did not change the cell viability. Both 100 μM and 400 μM BRZ in combination with 500 μM TMZ decreased the viability significantly compared to TMZ treatment alone (*n* = 3). * (Red): compared to Control group; # (Purple): compared to TMZ group; & (Orange): compared to ACZ 100 μM + TMZ group; % (Black): compared to BRZ 100 μM + TMZ group; $ (Green): compared to ACZ 400 μM + TMZ group. Results were obtained from 3 independent experiments. In (**A**–**C**,**E**–**G**), data are presented as mean ± SD, student’s *t*-test was used to analyze (**A**–**C**), One-way ANOVA was used to analyze (**E**–**G**), *, & *p* < 0.05; ## *p* < 0.01; ***, ###, &&&, $$$ *p* < 0.001, ns: not significant.

**Figure 6 ijms-23-00157-f006:**
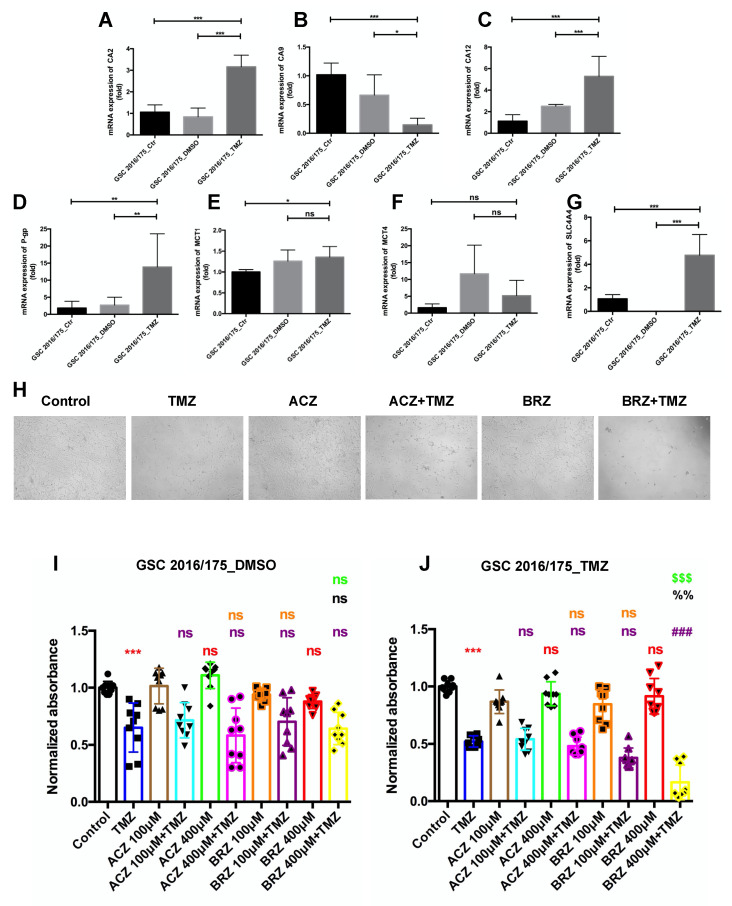
BRZ is more effective than ACZ in causing re-sensitization of long-term TMZ-resistant GSCs. (**A**–**G**) qPCR measurement for three GBM-related *CA* genes (*CA2*, *CA9*, and *CA12*), three monocarboxylate transporters (*MCT-1*, *MCT-4*, and *SLC4A4*), and drug-resistance gene (*P-gp*) which showed differential expression between TMZ resistant cell and its paired control/DMSO cells (*n* = 3). (**H**) Cell morphology changes after TMZ and CA inhibitor stimulation for 14 d. (**I**) Co-treatment with C50 dose (3 μM) of TMZ and 100 μM/400 μM ACZ or BRZ did not decrease the cell viability compared to single treatment with TMZ on DMSO control cells (*n* = 3). (**J**) Both 100 μM and 400 μM BRZ in combination with IC50 dose (250 μM) TMZ decreased the viability significantly compared to TMZ treatment alone on TMZ-resistant cells, however, ACZ in combination with TMZ did not change the cell viability (*n* = 3). * (Red): compared to Control group; # (Purple): compared to TMZ group; & (Orange): compared to ACZ 100 μM + TMZ group; % (Black): compared to BRZ100 μM + TMZ group; $ (Green): compared to ACZ 400 μM + TMZ group. Results were obtained from 3 independent experiments. Data are presented as mean ± SD, One-way ANOVA was used to analyze the data, * *p* < 0.05; **, %% *p* < 0.01; ***, ###, $$$ *p* < 0.001, ns: not significant.

**Figure 7 ijms-23-00157-f007:**
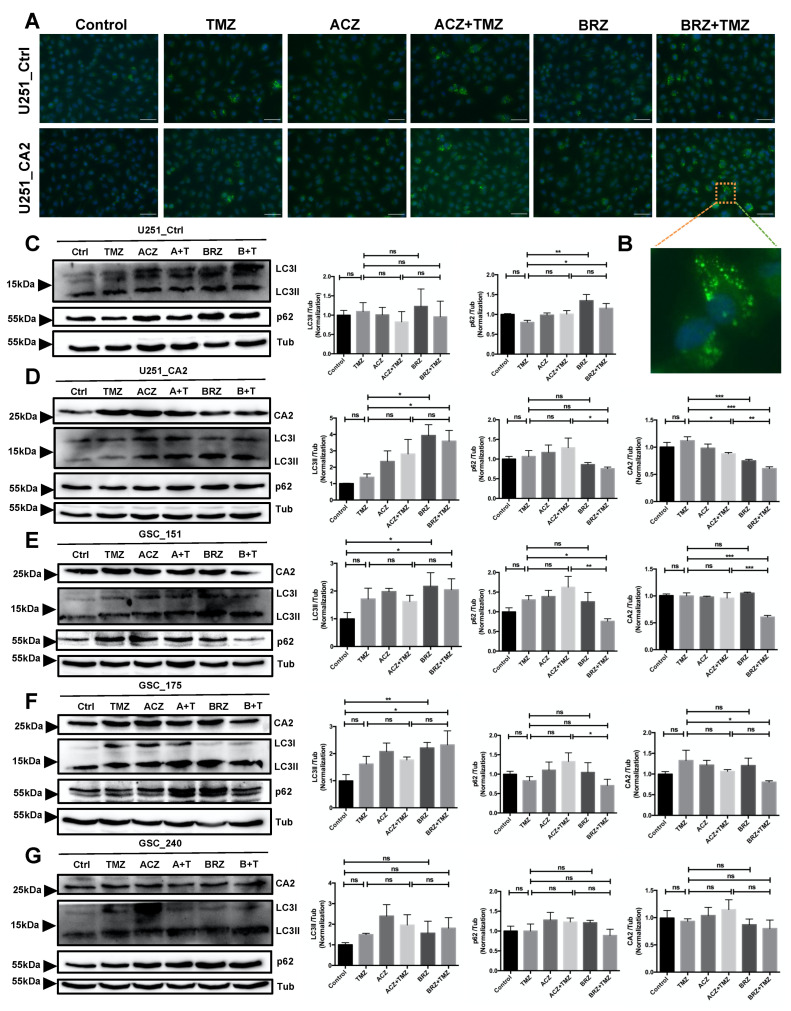
The combination of BRZ and TMZ increased cell death in GBM stem-like cells by activating autophagy (**A**,**B**) Autophagy marker LC3 immunostaining of U251_Ctrl and U251_CA2 cells after TMZ and ACZ/BRZ stimulation for 24 h. Co-treatment with TMZ and BRZ induced the expression of LC3 puncta in U251_CA2 cells (scale bar: 50 μm). (**C**,**D**) Western Blotting of autophagy-related proteins and CA2 protein in U251_Ctrl and U251_CA2 cells with the same treatment as in (**A**). TMZ plus BRZ did not increase the protein expression of LC3II in U251_Ctrl cells compared to TMZ treatment alone (**C**) but increased in U251_CA2 cells (**D**) (*n* = 3). (**E**–**G**) Western Blotting of autophagy-related proteins and CA2 protein in GBM stem cells with the same treatment as in Figure 5E–G for 24 h stimulation. Compared with the control group, TMZ monotherapy has a tendency to increase the protein level of LC3II, but the expression of it is significantly increased in BRZ alone and TMZ combined with BRZ treatment (*n* = 3). Results were obtained from three independent experiments. Data are presented as mean ± SEM, One-way ANOVA was used to analyze the data, * *p* < 0.05; ** *p* < 0.01; *** *p* < 0.001, ns: not significant.

## Data Availability

The raw data supporting the conclusions of this article will be made available by the authors, without undue reservation.

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
