# Peer review of "Inhibition of Carbonic Anhydrase 2 Overcomes Temozolomide Resistance in Glioblastoma Cells"

_ijms, 2021, doi:10.3390/ijms23010157_

Round 1

Reviewer 1 Report

In this study, the authors examined the role of CA II for TMZ therapy resistance in glioblastoma. Expression analyzes indicate an influence of CAII on TMZ treatment in GBM patients. The effects of stable CAII overexpression and/or inhibition with the carbonic anhydrase inhibitors ACZ and BRZ alone and in combination with TMZ were investigated in vitro using various cell systems and methods such as viability, invasion, OCR, ECAR, autophagy and expression analysis. This is an interesting extensive study to the importance of CAII for the therapy of glioblastomas. However, there are a few points that need to be addressed.

Major:
- The statements regarding to the specificity of the two CA inhibitors ACT and BRZ must be checked considering the Ki values of various carbonic anhydrases  such as I, II, IX, XII. 
-  Please show protein expression levels of CAII, CAIX and CAXII corresponding to mRNA data of figures 1e,f,g, and 6a,b,c.
- The authors should declare which experiments were technical replicas or/and independent experiments.

Minor:
- The mRNA levels and the transfection efficiency of the CA2-overexpressing cell lines should be supplemented.
- The statements in the legend of Figure 3 must be checked.
- TCGA data must be quoted.

Author Response

On behalf of all authors, I am delighted to submit the revision of our manuscript entitled, “Specific Inhibition of Carbonic Anhydrase 2 overcomes temozolomide resistance in glioblastoma cells” (Manuscript ID: ijms-1484897) to be considered for publication in International Journal of Molecular Sciences. All authors agree with the final content of the manuscript, and this work is not under consideration elsewhere. We thank the editor and the reviewers for their positive reply and the helpful comments. We are happy to fully address these in our revised manuscript. All changes in the revised manuscript are marked in red.

Point to Point response to reviewer’s comments

General comments to Reviewer 1:

Thanks for the constructive criticism. In response to the general comments by this reviewer, we tried to improve the introduction and the methods section by adding the relevant information. We hope that the additional information provided is sufficient to justify acceptance of our manuscript.

Reviewer 1:

In this study, the authors examined the role of CA II for TMZ therapy resistance in glioblastoma. Expression analyzes indicate an influence of CAII on TMZ treatment in GBM patients. The effects of stable CAII overexpression and/or inhibition with the carbonic anhydrase inhibitors ACZ and BRZ alone and in combination with TMZ were investigated in vitro using various cell systems and methods such as viability, invasion, OCR, ECAR, autophagy and expression analysis. This is an interesting extensive study to the importance of CAII for the therapy of glioblastomas. However, there are a few points that need to be addressed.

Q1: The statements regarding to the specificity of the two CA inhibitors ACZ and BRZ must checked considering the Ki values of various carbonic anhydrases such as I, II, IX and XII.

Responses: We have now included the inhibitory properties with ACZ and BRZ against some human (h) carbonic anhydrase isozymes in the revised manuscript in the Table S3 and S4. The cited publication was also included in the revised manuscript based on two reviews by Supuran. In one of these reviews, the Ki value for brinzolamide is similar when comparing hCA2 and hCA12, but the difference is that the Ki value for CA2 was determined with the full-length enzyme whereas the Ki value for CA12 was determined with the catalytic domain of CA12 only. For some carbonic anhydrases and compounds it was shown that the Ki values can differ by a factor of 30 when comparing catalytic domains with the full-length enzyme, suggesting that the Ki value for brinzolamide can be much higher than 3 nM for CA12 when the full-length enzyme is present. Nevertheless, we used brinzolamide as the “most” specific CA2 inhibitor, which is also shown in table S4.

Q2: Please show protein expression levels of CAII, CAIX and CAXII corresponding to mRNA data of figures 1e,f,g, and 6a,b,c.

Responses:

In the revised version, we included all western blot data for the CA2 expression in GBM cell lines, GBM stem cells and in TMZ resistant cells. These data are shown in Figure S1B and Figure S5. Sine we only provided comparative data on the expression levels of CA9 and CA12 to argue our focus on CA2, we feel that showing protein data on CA9 and 12 are out of the focus of this work and do not really provide novel information. In many publications it has been demonstrated that the mRNA levels correspond well with protein levels of CA9 and CA12. Also our hypoxia data in Figure S2 demonstrate the agreement with previous data providing evidence for hypoxia-dependent regulation of CA9.

Q3: The authors should declare which experiments were technical replicas or/and independent experiments.

Responses: The detailed replicas/independent experiments were now included in the corresponding figure legends in our revised manuscript.

Q4: The mRNA levels and the transfection efficiency of the CA2-expressing cell lines should be supplemented.

Responses: We appreciate the reviewer’s question. In fact, when we did the experiment, we used qPCR and Western Blot to evaluate the transfection efficiency. We are happy to add the CA2 mRNA expression data in the figure 2A (U87 cell) and figure 2C (U251 cell) of the revised manuscript.

Q5: The statement in the legend of Figure 3 must be checked.

Responses: We are sorry for the ambiguous legend of Figure 3.

The detailed legend of Figure 3 is now included in the revised manuscript.

Q6: TCGA data must be quoted.

Responses: We added a paragraph in the method section of the revised manuscript regarding the TCGA dataset in GBM.

Reviewer 2 Report

The current study entitled “Specific Inhibition of Carbonic Anhydrase 2 overcomes te-2 mozolomide resistance in glioblastoma cells” by Zhao et al has addressed a critical issue regarding the management of glioblastoma; the resistance to the first-line standard chemotherapy temozolomide (TMZ). The study outcomes revealed the potential carbonic anhydrase inhibitor-2 as a chemo-sensitizing drug target in recurrent glioblastoma, as well as, proposed a combined treatment of TMZ with specific carbonic anhydrase inhibitor-2 (brinzolamide) to tackle GBM chemoresistance and recurrence. The manuscript is well-organized and well-written, and the language of the manuscript is sound. The results are interesting and were well-discussed, and the data presented is compelling enough to warrant publication in Int. J. Mol. Sci.

Author Response

Thank you for this positive review, much appreciated. 

Round 2

Reviewer 1 Report

I am satisfied with the changes in the manuscript. However, doubts about the limited specificity of BRZ will remain.

Author Response

Thanks for this advice. We share the concerns and have now changed the wording accordingly in our revised manuscript and avoided "specific", even in the title it is now removed.

In the discussion, we also provided a statement (see lines 427-435) regarding the Ki values of brinzolamide to indicate this fact.